# Functional Cognitive Disorders (FCD): How Is Metacognition Involved?

**DOI:** 10.3390/brainsci11081082

**Published:** 2021-08-18

**Authors:** Andrew J. Larner

**Affiliations:** Cognitive Function Clinic, Walton Centre for Neurology and Neurosurgery, Liverpool L9 7LJ, UK; a.larner@thewaltoncentre.nhs.uk

**Keywords:** functional cognitive disorders, metacognition: metamemory, anosognosia

## Abstract

Functional cognitive disorders (FCD) have become a subject of increasing clinical interest in recent years, in part because of their high prevalence amongst patients attending dedicated memory clinics. Empirical understanding of FCD based on observational studies is growing, suggesting a relationship to other functional neurological disorders (FND) based on shared phenomenology. However, understanding of FCD at the theoretical level has been lacking. One suggestion has been that FCD are disorders of metacognition, most usually of metamemory. In this article, a brief overview of these constructs is presented along with existing evidence for their impairment in FCD. Previous adaptations of theoretical models of FND to accommodate FCD are reviewed. A novel application to FCD of Nelson and Narens’ monitoring and control model of metamemory is then attempted, positing an improper setting of the monitoring function, with examples of ecological relevance. Formulation of FCD in light of a metacognitive model of anosognosia is also considered. Although lacking mechanistic and neuroanatomical sophistication, this metacognitive formulation of FCD may give pointers for future hypothesis-driven research and a pragmatic basis for management strategies.

## 1. Introduction: Functional Cognitive Disorders (FCD)

Functional neurological disorders (FND) have become a subject of increasing clinical and research interest in recent years [1]. The “functional” nomenclature is apparently the least offensive to patients with these conditions [2]. However, the shortcomings of this terminology have long been noted. For example, the Irish psychiatrist Maurice O’Connor Drury (1907–1976), who initially studied philosophy at Cambridge as a pupil, and later a friend, of Ludwig Wittgenstein (1889–1951), noted in his 1973 book entitled *The danger of words*, that:
“It has been suggested … that the word ‘functional’ should be used instead of ‘hysterical’. This … is an example of the fallacy of Molière’s physician [in the play *Le Malade imaginaire* (*The Imaginary Invalid*) of 1673] in that it pretends to explain by a learned circumlocution a condition which to date neither doctors nor patients understand”.([3], p. 270)
More recent commentators have also opined that “functional” is “no more enlightening” than other options, although serviceable and “accommodating in theoretical terms”.([4], p. 3496)

Interest in functional disorders has encompassed the domain of cognitive, as well as sensory and motor, function (Drury noted, in an unpublished series of lectures on hypnosis, that “there are a variety of ‘functional nervous disorders’” ([3], p. 398)). Previously denoted by a variable and extensive nomenclature [5], the coinage of the term “functional cognitive disorders” (FCD) by Stone et al. [6] in 2015 has helped to focus attention in this particular area. To what extent are FCD now understood? This might be considered from both empirical and theoretical standpoints. 

Empirically, the key clinical feature of FCD, as in other FND, is a discrepancy or internal inconsistency between subjective reports and objective measures, specifically in this instance of cognitive performance, most frequently memory performance. This is perhaps best encapsulated by the observation that patients are often able, during the history-taking component of clinical assessment, to remember what they claim to have forgotten and hence are able to give an account, often detailed, of these memory lapses [7]. The contrast with an erstwhile excellent or brilliant memory function is often complained of. Addressing the ontological challenge as to whether or not these entities have persisted over time, the incongruence between subjective symptoms and objective functional preservation may be detectable in historical accounts of memory symptoms [8], suggesting that FCD are not new disorders.

FCD may be viewed phenomenologically as the mirror image or antithesis of the situation encountered in some patients with Alzheimer’s disease who minimize, disclaim, or “deny” any memory defect, whilst their (often anguished) relatives give a very different account, and in whom objective cognitive testing confirms evidence of amnesia. This discrepancy, the reverse of FCD, is sometimes denoted as cognitive anosognosia (i.e., unawareness and denial of the amnesic deficit) or anosodiaphoria (lack of concern over the deficit), which may take a number of forms [9].

An increasing number of observational studies of FCD have been published (e.g., [7,10,11,12,13,14,15,16,17]; see [18,19,20] for reviews). These suggest that FCD are commonly encountered in dedicated memory and cognitive disorders clinics, sometimes accounting for more than half of the patients seen, typically in individuals younger than those with dementia and mild cognitive impairment (MCI). Clinical signs suggestive of the diagnosis of FCD include the “attended alone” sign (failure to bring a knowledgeable informant to the clinical consultation to provide collateral history despite the request to do so in the clinic appointment details), which has a high positive predictive value for the absence of any cognitive impairment [21], and the production of a written list of symptoms (*la maladie du petit papier*) [22]. Performance on cognitive testing, either simple screening tests or more detailed neuropsychological assessment, is relatively well preserved. Disturbance of mood [15] and of sleep [23] is common in these patients. Prognosis is uncertain, with some evidence that symptoms persist in the short [24] and long [25] term without progression, although possible misdiagnosis of early-stage neurodegeneration as FCD is a recognized pitfall [16,26] (for example, in a patient with an unusually high baseline, or because the objective testing was insufficiently sensitive). A positive family history of dementia, usually of the late-onset type, may be more common in FCD patients than in non-functional cognitive disorders [14]. Various FCD typologies have been suggested, including mood disorder, other functional disorders, medication effects, dementia health anxiety, normal cognitive experience, dissociative amnesia, and malingering [6]. 

The development of a consensus definition of FCD [27], superseding listings of core clinical features used to guide diagnosis hitherto [18], may facilitate further understanding of FCD if widely accepted and adopted (Box 1). For example, the suggested FCD typologies [6] might be examined using hierarchical cluster analysis. It may be that FCD, like psychiatric disorders, are characterized by a multitude of symptoms that may be present in different combinations, and hence may best be classified as heterogeneous syndromes rather than categorical disorders. 

Box 1Proposed operational definition of FCD (based on [27]).1. One or more symptoms of impaired cognitive function.2. Clinical evidence of internal inconsistency. ^a^3. Symptoms or deficit that are not better explained by another medical or mental disorder. ^b^4. Symptoms or deficit that cause clinically significant distress or impairment in social, occupational, or other important areas of functioning or warrants medical evaluation. ^c^ a. Internal inconsistency indicates a worse performance self-reported compared to objective evidence, or inconsistency between situations, or at different time points (i.e., variability over time, not stability nor a pattern of decline).b. Patients may have comorbid medical or psychiatric disorders as well as FCD.c. A minimum of 6 months duration should be considered. SpecifierSpecify if: with/without a linked co-morbidity.

The possible relationship to, or overlap with, other FND [6,15] is of particular interest, raising the possibility that FCD may be viewed as “functional neurological disorder—cognitive subtype”. Based on a systematic review of reported cognitive abnormalities in non-cognitive FND, fibromyalgia, and chronic fatigue syndrome, Teodoro et al. [28] proposed a unifying theory linking the observed deficits in aspects of attention, information processing, and vulnerability to distraction, with the proposed cognitive abnormalities in FCD. Some preliminary evidence in support of this theory has been presented, finding a shared profile of cognitive deficits in fibromyalgia and FCD [29]. Just as many psychiatric disorders, hitherto defined by the classical medical model of categorical diagnostic classification, appear to share an underlying cause, a broad susceptibility to mental health symptoms, the so-called p-factor [30], so it will be interesting to learn if functional disorders may also share an underlying cause or susceptibility, tentatively dubbed the f-factor [31].

Even with this empirical information, Drury’s admonition concerning the understanding of “functional” disorders may still stand, in that a theoretical explanation is lacking. One attempt to conceptualize FCD has posited that these may be disorders, indeed primary disorders, of metacognition, in particular of metamemory [32,33]. Before presenting the empirical evidence base in support of this possibility (Section 3) and exploring some possible theoretical models (Section 4 and Section 5), some brief comments about metamemory and metacognition are in order.

## 2. Metamemory and Metacognition

The term “metamemory” was first used by the developmental psychologist John Flavell (1971) to describe knowledge about one’s memory processes and contents [34]. Flavell later (1976) broadened the concept to encompass “metacognition” as referring to “one’s knowledge concerning one’s own cognitive processes and products or anything related to them” ([35], p. 232). Of course, considerations of “knowing about knowing” long predate these neologisms, going back as far as classical antiquity, specifically to certain of the works of Aristotle. In the sphere of cognitive psychology, Hart (1965) developed the idea of “feeling of knowing” (FOK) [36], which has been used as a method of assessing metamemory judgments for already acquired material. Judgments of learning (JOLs) form another category of postdiction used to assess (recallable) information following acquisition, whilst ease of learning (EOL) is a prediction before acquisition. These paradigms have been widely used to inform metamemory and metacognitive models in various disciplines, including not only developmental and adult psychology but also social psychology, educational psychology, in legal contexts, and neuropsychology.

Clearly metamemory, like memory per se, is a multidimensional, rather than a unidimensional construct, comprising many characteristics [37]. Various frameworks and taxonomies of metamemory and metacognition have been proposed, a simplified composite of which is presented in Figure 1, which deliberately overlaps with simplified taxonomies of memory processes. Whether metacognitive resources are domain-specific or domain-general (i.e., whether there is a g factor for metacognition) continues to be a subject of investigation [38].

Whilst metacognitive errors are probably not uncommon in everyday life, such as overconfidence in one’s own abilities, this may seldom result in presentation to medical services with persistent and distressing subjective cognitive complaints, which might justifiably be termed metacognitive disorder. From a clinical standpoint, a number of clinician-administered performance scales have been developed and used on occasion to assess metamemory, including but not limited to the Metamemory in Adulthood (MIA) Questionnaire [39], the Memory Functioning Questionnaire (MFQ) [40], and the Sehulster Memory Scale [41]. MIA and MFQ show convergent validity around a factor labelled memory self-efficacy (MSE) [42], defined as beliefs about capability to use memory effectively. However, “global” measures such as these have been criticized, and the use of measures of “local” metacognitive sensitivity, tracking changes in moment-to-moment cognitive performance, has been advocated [33]. 

Metacognition, particularly metamemory, has been examined in the context of a number of neurological disorders [43], including Alzheimer’s disease (AD) [44,45,46], frontotemporal dementia [47], and multiple sclerosis [48]. Metamemory tasks have also been used to explore the anosognosia of AD [49,50]. In each of these conditions, metacognitive deficits are only one element in a broader landscape of cognitive dysfunction. Whether primary clinical disorders of metacognition exist remains unresolved, but FCD may be a candidate.

## 3. FCD: Evidence for Impaired Metacognition

What empirical support exists for the possible conceptualization of FCD as disorders of metacognition, in particular of metamemory? Currently this evidence base is limited.

Using subscales of the MIA, Metternich et al. found lower memory self-efficacy in patients diagnosed with functional memory disorder than in healthy controls despite no significant differences in objective cognitive performance [51].

The Subjective Memory Complaint (SMC) Likert Scale [52], in which patients self-rate their memory on a five-point Likert scale ranging from poor to excellent, has also been used as a global measure of metacognitive performance. In patients self-rating their memory as either poor or fair, and hence classified as SMC+, the sensitivity for a diagnosis of FCD/no cognitive impairment was good (0.87) although the positive predictive value was suboptimal (0.57). This suggested the test may be adequate for ruling in a diagnosis of FCD [32], although the SMC Likert Scale did not appear to have differential diagnostic value for distinguishing dementia and non-dementia cases [53].

The possibility that the direction of causality runs from misdirected attention to metacognitive deficits, rather than metacognitive deficits being primary in FCD, has not been entirely discounted by the empirical data [33]. An important therapeutic corollary of the exact direction of causality is whether or not interventions directed to metamemory processes could affect memory symptoms in FCD.

## 4. FND Models and Their Possible Relation to FCD

The possible relationship of FCD to other functional neurological disorders (FND) suggests that looking at existing models and theoretical frameworks for FND might inform conceptions of FCD. Whilst many psychological theories for FND have been mooted (see [54] for a concise summary), few appear to have been specifically adapted as possible explanations for FCD.

### 4.1. Integrative Cognitive Model (ICM)

The integrative cognitive model (ICM) of psychogenic non-epileptic seizures (PNES), another FND, was developed by Brown and Reuber [55], building upon a prior model of medically unexplained symptoms [56]. The ICM was subsequently drawn upon by Bhome et al. [33] as a model for FCD. This adaptation substituted a “cognitive scaffold”, comprising metacognitive abilities, for the “seizure scaffold” of the original ICM and acknowledged the distinction between the persistent symptoms of FCD compared to the episodic nature of PNES. Bhome et al. suggested that memory lapses due to misdirected attention might activate the cognitive scaffold, likewise emotional responses to such lapses and pre-existing stress [33].

### 4.2. Bayesian Model

A hierarchical Bayesian model of functional motor and sensory symptoms was developed by Edwards et al. [4] based on the concept of predictive coding in the nervous system [57]. The authors posited a discrepancy between the “top down” predictions of motor and sensory processes (which may also be conceptualized as abnormal beliefs, expectations, or priors), and the “bottom up” interpretation of sensory inputs, in which explicit attention to the relevant body part amplified, or even caused, the symptomatology. The neurobiological mechanisms that might underpin the Bayesian minimization of prediction errors include changing synaptic activity and connection strengths. Synaptic neuromodulation, with high synaptic gain mediated by attention, might provide a mechanistic explanation of the functional symptoms, and hence give biological plausibility to this model [4]. Bhome et al. [33] suggested this model could be adapted to FCD, viewed as an impaired metacognitive ability causing disparity between expectations (“abnormal priors”) of cognitive ability and actual performance.

### 4.3. Unifying Theory Model

As part of their unifying theory linking deficits in attention, information processing, and vulnerability to distraction in FND, fibromyalgia, and chronic fatigue syndrome with the proposed cognitive abnormalities in FCD, Teodoro et al. proposed a mechanistic model for cognitive difficulties across the spectrum of FCD [28]. Therein, excessive interoceptive attention was postulated to reduce attentional reserve and hence externally directed attention, leading to increased susceptibility to distraction and slow information processing. Heightened self-monitoring of cognitive processes might lead to over-interpretation of attentional lapses, perhaps in the context of memory perfectionism, and hence to subjective memory complaints. 

## 5. Metacognitive Models and Their Possible Relation to FCD

Dixon ([36], p. 49) noted that “understanding metamemory may help researchers and clinicians to identify and understand … the occasional gaps between hypothetical memory competence and actual memory performance”. None of the aforementioned models of FND as originally presented [4,28,55] specifically referred to models of metamemory per se, although some potential adaptations applicable to FCD have subsequently been considered [33]. Appeal to existing models and theoretical frameworks of metacognition might inform conceptions of FCD. Metacognitive models of anosognosia may also be pertinent to this exploration. 

### 5.1. Metamemory “Monitoring and Control” Model of Nelson and Narens

A significant theoretical development in the field of metamemory was Nelson and Narens’ (1990) distinction between monitoring and control processes [58]. Following the usage of David Hilbert (“metamathematics”) and of Rudolf Carnap (“metalanguage”), Nelson and Narens proposed that cognitive processes be split into two interrelated levels, the object-level and the meta-level, wherein the latter contained a “dynamic model (i.e., a mental simulation)” of the former. Flow of information between the two levels was defined by two “dominance relations”, specifically control and monitoring, whereby the meta-level monitors object-level cognitions and thus controls or modifies unstable object-level processes (Figure 2). As the two mechanisms interact at any given moment, this may be described as an online supervisory or executive system.

This two-level formulation was also implicit in Flavell’s earlier characterization of metacognition as referring to “the active monitoring and consequent regulation and orchestration of these processes in relation to the cognitive objects or data on which they bear” ([35], p. 232). Hence, this monitoring and control formulation essentially constitutes a feedback loop model, with a comparator function assigned to the meta-level. 

As such, this metacognitive model is easily transposed to a theoretical model of the neuropsychological mechanisms of various forms of anosognosia that was proposed by Heilman [59] (apparently independent of any knowledge of the work of Nelson and Narens). This may be a particularly pertinent comparison if FCD are the “mirror image” or reverse of cognitive anosognosia. In Heilman’s formulation, “neural representations” receive feedback from, and hence monitor, a behavioural output or effector level, permitting comparison between output and representation in order to detect matching or mismatching. Heilman’s neural representation and behavioural output may thus be analogous to the meta-level and object-level, respectively, of Nelson and Narens’ metamemory framework (Figure 2).

Heilman proposed that defects in the comparator or monitoring systems may result in clinical syndromes. For example, sensory (Wernicke-type) aphasia was conceptualized as a consequence of destruction (e.g., by stroke) of the neural representations of words, such that monitoring of behavioural (speech) output could not occur because there were no intact neural representations. Destruction of a monitoring system was also postulated to be one of the mechanisms accounting for the syndrome of cortical blindness (Anton’s syndrome) in which patients deny their loss of vision, along with “false feedback” to the monitor. This latter might be from visual inputs mediated via subcortical visual pathways, or from visual imagery that is misinterpreted as coming through visual input, hence accounting for the visual hallucinations that may occur in Anton’s syndrome (a similar mechanism, but without anosognosia, may underlie the visual hallucinations of Charles Bonnet syndrome). Although not specifically addressing the cognitive anosognosia of Alzheimer’s disease, Heilman did suggest that denial of amnesia might also be a consequence of “false feedback” to a monitoring system through the retrieval of existing memories, even if they were incorrect ([59], p. 57).

In all, Heilman suggested four potential mechanisms that might account for anosognosia: impaired monitor; absence of feedback; false feedback; and improper setting of the monitor ([59], p. 53). These mechanisms were not seen as mutually exclusive. For example, in denial of hemiplegia, typically following right hemisphere stroke, there may not only be a failure of feedback as a consequence of sensory (proprioceptive) defects in the hemiplegic limb but also a “feed forward” or “intentional” component: failure of an expectation of movement (neural representation) to be fed into the comparator might result in an absence of mismatch in an intact monitor when no feedback (behavioural output) occurred, with the resulting clinical correlate of denial of hemiplegia.

### 5.2. Proposed Adaptation of the Nelson and Narens’ Metamemory Model to FCD

In light of Nelson and Narens’ model of metamemory [58] and Heilman’s proposed neuropsychological mechanisms of failure of monitoring [59], FCD might be conceptualized theoretically as a consequence of defective monitoring or comparator function within a metacognitive feedback loop (here “monitoring” is being used to denote a process of updating inputs with the feedback from outputs, and not to suggest some form of introspection, or “inner sense”). This might come about in various ways, some more likely than others. As no structural neuroimaging or neuropathological correlate of FCD has been yet reported, it may be presumed that the monitor (meta-level, or neural representation) is likely to be intact, and likewise that there is no absence of, or false, feedback. Improper setting of the monitor would therefore seem the most likely mechanism. 

How is the meta-level comparator for cognitive or memory functions set, the neural representations which determine memory self-efficacy? As suggested by Herzog, commonly held negative implicit theories about ageing and memory influence perceptions of memory functioning and act as the lens though which experiences of remembering and forgetting are filtered. Moreover, this expectation of impending cognitive decline may be greater in younger than in older adults [60]. Furthermore, overconfidence about preceding memory performance is a common finding [61], which may lead to unrealistic setting or calibration of the monitor or comparator function. 

Memory lapses and slips are a frequent occurrence in everyday life [62,63]. Monitoring of this behavioural output from the object-level through feedback to the meta-level neural representations might result in a mismatch with (feed forward) expectations about memory performance. Indeed, this expectation may mean that memory slips are over-monitored or over-attended to, exacerbating mismatch by means of a vicious circle (positive feedback?), with the resulting clinical correlate of subjective memory complaints and presentation to medical services. A positive family history of dementia, previously characterized as a factor sensitizing to memory symptoms [64], may be seen as influencing the meta-level monitor, as may memory perfectionism [28].

In the interest of ecological validity, it may be worthwhile to consider this model of FCD as a primary disorder of metamemory in the context of two of the most familiar complaints voiced by patients presenting to memory clinics when they are invited by the clinician to “give some examples of how your memory lets you down”: forgetting the names of familiar people; and forgetting words or what they were going to say in conversation with others. The prevalence of these complaints is not simply age-related: a study using the Subjective Memory Complaints scale (not to be confused with the SMC Likert Scale [52]) found that items addressing these symptoms (“Do you ever forget names of family members or friends?” and “Do you ever have difficulties in finding particular words?”) were endorsed no differently in young and old healthy participants [65]. 

Learning people’s names is an example of cross-modal, non-contextual, paired-associate learning. Hence, this is material about which, once acquisition has occurred, FOK judgments can be made [36]. However, lack of meaningful, semantic associations renders the learning of people’s names harder than that of other nouns, which may explain why this symptom is common even in healthy individuals [66]. Unrealistic or overoptimistic setting of the meta-level monitor function may result in a mismatch with the output of memory performance for the challenging task of recalling personal names. 

Forgetting what one is going to say, often referred to as losing the thread of one’s conversation, is an example of the capacity limitations of the online working memory system, and hence may be a function of attentional rather than mnemonic mechanisms. The online nature of the monitoring and control processes in Nelson and Narens’ metamemory model may be of significance here. In addition, the intentional, feed forward, prospective nature of what one is going to say might lead to a mismatch with the output if the monitoring expectations are set too high or the focus of attention is misdirected. This might also be an example of the blocking phenomena encountered when switching from implicit to explicit processing, self-focused attention leading to a deterioration in performance as in other FND [4]. Change in focus of attention may also underpin impaired performance in so-called “performance validity” tests [67].

### 5.3. Other Metacognitive Models, including the Cognitive Awareness Model (CAM)

The previous considerations regarding the proposed adaptation of Nelson and Narens’ metamemory model to FCD are compatible with other suggested formulations of metacognition, for example, the “8 pillars of metacognition” model in which the capacities of monitoring, regulation, and adaptation of cognitive mechanisms are central [68]. 

More directly relevant to FCD is the Cognitive Awareness Model (CAM) [69] and its reformulations [70,71], which were developed by Morris and his colleagues to explain anosognosia in Alzheimer’s disease (Heilman’s formulation of anosognosia did not have recourse to ideas about metacognition and, as previously mentioned, did not specifically address cognitive anosognosia [59]). Although more complex than Nelson and Narens’ metamemory model [58], as required to explain the heterogeneity of anosognosia [9], the essential feature of CAM is that task performance is monitored by comparator mechanisms. Specifically, a module named Cognitive Comparator Mechanisms (CCM) detects task failure when compared to an existing Personal Database, which may then be updated, and which feeds into a Metacognitive Awareness System (MAS) that “provides a phenomenological sense of awareness of failure” ([71], p. 1560). Deficits of awareness may be associated with impairment at various levels of CAM. By extension, FCD might also be explicable using the CAM, for example, as a defect in CCM, such that normal memory slips are detected as memory failures and, hence, via the MAS, as (inappropriate) awareness of failure. (It should be noted that Morris and Mograbi ([71], p. 1561) use “meta-level” in a different sense to that of Nelson and Narens [58].)

### 5.4. Metamemory Model Limitations and Comparisons with FND Models

There are, of course, shortcomings to the proposed theoretical metacognitive formulation of FCD. Identical to the limitations of the ICM model, as acknowledged by its authors [55], the proposed metamemory model of FCD adapted from Nelson and Narens lacks mechanistic precision and does not map to anatomical structures. Nevertheless, the “remarkable consensus on the heuristic value of [the] … important process theory of Nelson and Narens” ([72], p. 17) has encouraged this exploration of its potential applicability to FCD. 

Clearly the proposed metamemory model of FCD has areas of overlap with the FND models when they are extrapolated to the cognitive domain (Section 4), as might be expected since many of these models are using the same or similar concepts but with differing terminology. The parsimonious formulation of FCD as “functional neurological disorder—cognitive subtype” might be anticipated to imply mechanistic overlap between FND and metamemory models of FCD, hence underpinning not only sensory and motor but also cognitive symptoms. For example, the previous adaptation of the ICM to FCD [33] might be taken further by suggesting that “shaping factors” for the cognitive scaffold, analogous to those shaping the “seizure scaffold”, might include “hard-wired behavioural tendencies”, such as beliefs about age-related memory function and dysfunction, and “experiences misinterpreted” as memory loss/amnesia. 

Although the proposed FCD metamemory model is not hierarchical like the Bayesian model of functional motor and sensory symptoms [4], it does, as per Nelson and Narens’ metamemory model [58], make predictions about task performance and hence might correspond to the prediction and prediction error units in each individual level of the Bayesian model ([4], p. 3497, Figure 1). The latter model’s “abnormal priors” might be equated with the impaired monitoring function of the meta-level comparator in the metamemory model, in terms of unrealistic memory expectations, wherein overly precise expectations of memory performance override “bottom up” sensory data.

An inferential “Bayes optimal” hierarchical system, as envisaged for perception, decision making, and sensorimotor integration [73], posits a single mechanism [4], the features of which may generalize from one system (sensory, motor) to another (cognitive). Overweighting of prior beliefs over sensory data is a common phenomenon in predictive coding networks and may underpin not only perception of optical illusions and the placebo and nocebo effects but also the subjective/objective incongruence of FND, including FCD. Alternatively, it may be that brain function in FCD patients may be viewed as Bayes “non-optimal”, with failure to update priors adequately on the basis of sensory feedback. In the unifying theory model [28], the proposed memory perfectionism and heightened self-monitoring of cognitive processes may play a similar role in (mis-)calibrating the priors.

The proposed FCD metamemory model may be too reductive, too simplistic, and hence not able to account for or to accommodate the possible heterogeneity of FCD [6]. The frequency of mood disorders such as depression in FCD [33] might be seen to undermine the idea of FCD as a primary disorder of metamemory. However, metacognitive mechanisms have also been postulated in the occurrence of depression [74]. Another clinical observation that is problematic for the model relates to instances of acute onset of FCD, such as the dissociative amnesia type, which implies a sudden rather than gradual change in synaptic neuromodulation.

## 6. Summary: FCD, Anosognosia, and Wittgenstein

The suggested metacognitive formulations of FCD and anosognosia might be summarized, in the spirit of Wittgenstein, by the construction of a (modified!) truth table (after *Tractatus Logico-Philosophicus* §4.31) as a useful heuristic. Herein “true” and “false” are not being used as per the Wittgensteinian reportable propositions, but merely to indicate functional status, normal or abnormal, without further specification of the nature of the dysfunction. This binary classification is used here for simplicity, as it is likely that these functions are continuous rather than categorical.

The particular case considered relates to memory expectations (variously denoted: meta-level; feed-forward; top-down; descending; intentional) and memory performance (object-level; feed-back; bottom-up; ascending; actual). The key point is whether mismatch (discrepancy; inconsistency; incongruence) or no mismatch is detected by the monitoring comparator function, with the resulting inferences (or “phenomenological sense” [71]), their relation to the clinical context, and the consequent behavioural outcomes forming the substance of clinical practice (Table 1).

## 7. Future Prospects and Concluding Remarks

Some attempts have previously been made to accommodate the conceptualization of FCD as primary disorders of metacognition, principally of metamemory, within existing models of FND. Herein, it has been shown that FCD might also be accommodated within the influential Nelson and Narens’ theoretical framework of metamemory. Specifically, it has been proposed, based on analogy with models of anosognosia, that improper setting of the monitoring or comparator function at the meta-level of the framework leads to a mismatch with the output from the object-level feedback, and with the clinical correlate of the subjective memory complaint, despite a relatively preserved objective memory function. 

Although this proposal of dysfunction within a metacognitive feedback loop lacks mechanistic precision, nevertheless this formulation may give pointers for future hypothesis-driven research in FCD and a pragmatic basis for management strategies. 

Aided by an operationalized diagnostic definition of FCD [27], research studies of the neural substrates of metacognition and metamemory in FCD patients, for example, using “local” measures of metacognition and functional imaging techniques, are indicated [33]. It would be interesting to know if these neural substrates overlapped in any way with those previously defined in the anosognosia of Alzheimer’s disease [75]. Such studies might also clarify the relation between FCD and other FND and hence whether or not the idea of an underlying f factor is credible or whether fractionation into particular typologies is more appropriate. This might entail the development of a new, more descriptive terminology (perhaps “metamnesia”, or “metamnemonic disorder”?).

At the therapeutic level, the model has clear implications for management of FCD. To adopt a computing analogy, it envisages FCD as a disorder of memory software, in contrast to disorders of memory hardware such as Alzheimer’s disease and Korsakoff’s syndrome. Since the memory monitoring function is conceptualized to be dynamic, it should be susceptible to “recalibration” or “resetting”, as occurs physiologically. Methods to do this, which might be explored in randomized controlled trials, might include modalities such as cognitive behavioural therapy and computerized metacognitive training, or even regular meditation [76].

It is to be hoped that with increasing attention and research, FCD will become better understood and not merely, as Drury may have feared, “a learned circumlocution” ([3], p. 270).

## Figures and Tables

**Figure 1 brainsci-11-01082-f001:**
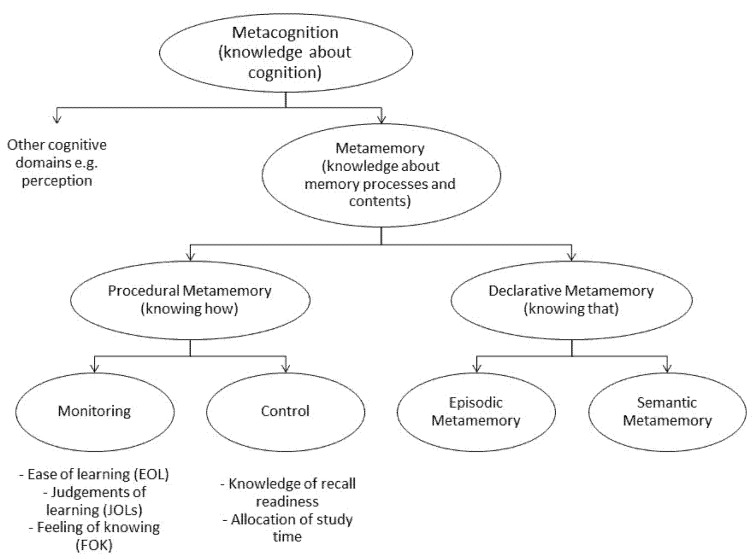
A possible taxonomy of metacognition.

**Figure 2 brainsci-11-01082-f002:**
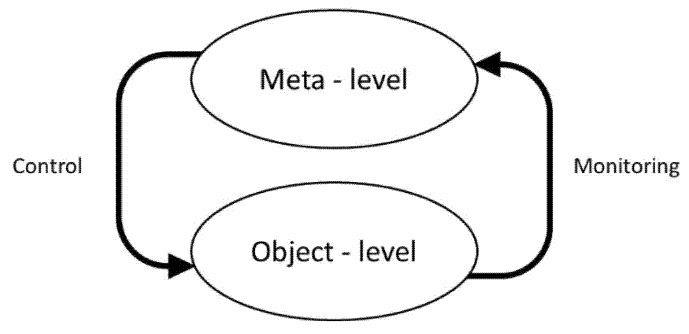
Nelson and Narens’ theoretical framework for metamemory (adapted from [58]).

**Table 1 brainsci-11-01082-t001:** “Truth table” for metamemory models of FCD and anosognosia.

Memory Expectations	Memory Performance	Comparator	Inference or Phenomenological Sense	Relation of Inference to Clinical Context	Behavioural Outcome
T	T	No mismatch	“Everything’s OK”	Appropriate (Inference true)	Well and aware
T	F	Mismatch (=prediction error)	“Something’s wrong”	Appropriate (Inference true)	Unwell and aware = amnesia
F	T	Mismatch (=prediction error)	“Something’s wrong”	Inappropriate (Inference false)	Well and unaware = FCD
F	F	No mismatch	“Everything’s OK”	Inappropriate (Inference false)	Unwell and unaware = anosognosia

## Data Availability

Not applicable.

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
