# Peer review of "Functional Cognitive Disorders (FCD): How Is Metacognition Involved?"

_brainsci, 2021, doi:10.3390/brainsci11081082_

Round 1

Reviewer 1 Report

This is a very useful overview  of different theoretical approaches to understanding FCD, followed by a proposition that the Metacognitive model might be used instead. My only main criticism is this could be framed in a way that highlights how many of these models are using the same or similar concepts but with differing terminology.

A second point to consider: is it a problem for all these various models, which may deal in synaptic neuromodulation (synaptic plasticity building up each time an error occurs), that FNDs (including FCD) sometimes appear suddenly, rather than build up gradually over time?

A few minor suggestions, with line markers:

The initial section of the Introduction (23-40) felt disjointed and could be edited to ensure each paragraph makes a clear point.

47: here you might like to make mention of internal inconsistency. 

189-190: consider further punctuation to make it clear abnormal beliefs/ expectations/ priors/ top down predictions all refer (broadly) to one thing.

197-199: I don't quite understand how (if?) dissociative amnesia links in here, and whether it is a paragraph by itself (with the sentence starting on line 199 being its own paragraph or part of the paragraph above).

256: is the "false feedback" you refer to analogous to Charles Bonnet syndrome? If so it might aid understanding to mention this syndrome.

318: the word "occasion" here is confusing me, do you mean cause, or lead to?

371-373: "high sensitivity will generate false positives" you might want to clarify the meaning of sensitivity (test sensitivity vs having a sensitive personality). I think I may have misunderstood how this situation relates the the remainder of the paragraph, please could you expand to clarify?

Author Response

Reviewer 1:

This is a very useful overview of different theoretical approaches to understanding FCD, followed by a proposition that the Metacognitive model might be used instead. My only main criticism is this could be framed in a way that highlights how many of these models are using the same or similar concepts but with differing terminology.

Author response: This point has now been added (Section 5.3, paragraph 2).

A second point to consider: is it a problem for all these various models, which may deal in synaptic neuromodulation (synaptic plasticity building up each time an error occurs), that FNDs (including FCD) sometimes appear suddenly, rather than build up gradually over time?

Author response: This point has now been discussed (Section 5.3, paragraph 5).  

The initial section of the Introduction (23-40) felt disjointed and could be edited to ensure each paragraph makes a clear point.

Author response: These lines have been re-ordered in the hope that each point is now clearer (Section 1, paragraphs 1 and 2).

47: here you might like to make mention of internal inconsistency. 

Author response: This suggestion has been adopted (section 1, paragraph 3).

189-190: consider further punctuation to make it clear abnormal beliefs/ expectations/ priors/ top down predictions all refer (broadly) to one thing.

Author response: These lines have been rewritten in light of this comment in the hope that their meaning is now clearer (Section 4.2, paragraph 1).

197-199: I don't quite understand how (if?) dissociative amnesia links in here, and whether it is a paragraph by itself (with the sentence starting on line 199 being its own paragraph or part of the paragraph above).

Author response: Sorry for any misunderstanding, the sentence mentioning dissociative amnesia has now been omitted.

256: is the "false feedback" you refer to analogous to Charles Bonnet syndrome? If so it might aid understanding to mention this syndrome.

Author response: This sentence has been developed to encompass this idea (Section 5.1, paragraph 4).

318: the word "occasion" here is confusing me, do you mean cause, or lead to?

Author response: Now changed to read “lead to” (Section 5.1, paragraph 11).

371-373: "high sensitivity will generate false positives" you might want to clarify the meaning of sensitivity (test sensitivity vs having a sensitive personality). I think I may have misunderstood how this situation relates the the remainder of the paragraph, please could you expand to clarify?

Author response: Sorry for any misunderstanding, this sentence has now been omitted.  

Reviewer 2 Report

This article raises the intriguing hypothesis that ‘Functional Cognitive Disorder’ is ‘primary disorder of metacognition’ (p. 4, line 149). As such, this is an interesting hypothesis. However, I admit I found the manuscript hard to follow. It felt that the author tried to cover too much, and the end-result is a product that is too broad and not sufficiently in-depth. I had a hard time noticing how the following attempt advances our understanding beyond Bhome et al (2019)’s review on the same topic. I’m sorry I do not have a more positive disposition toward it. Below I list some observation that might be useful to the author.

As described by the authors, ‘Functional Cognitive Disorder’ refers to a subjective deficit in cognition in the context of objectively normal performance. An illustrative example would be a person who seeks the services of a memory clinic or a cognitive neurology clinic despite having a normal neuropsychological profile.

First, I want to raise as an important caveat that the gap between subjective and objective deficit does not necessarily mean that the person has a deficit (in metacognitive) judgment. Instead, the patient might very well be impaired relative to his/her baseline and the cognitive battery fail to detect such a decline, either because the patients’ baseline was unusually high, or because the objective test is less sensitive than the patient’s insight.

Other additional criteria have been proposed to better describe Functional Cognitive Disorders. While the author does cite such work (e.g., by McWhither et al; Stone et al, 2015, etc.) as a reader I had a hard time getting a solid grasp of the diagnosis criteria from the text. . For example, it was unclear whether the diagnosis requires a failure to progress (so to differentiate it from dementias). The author would be well advised to start the article with a clear definition of FCD, such as “Functional cognitive disorder (FCD) is characterised by the experience of persistent and distressing subjective cognitive difficulties in the absence of detectable objective cognitive deficit and underlying brain pathology” (Bhome et al., 2019)

Putting these issues aside, it is definitional that people who exhibit a gap between   objective performance and the subjective self-assessment, are committing a metacognitive error. However, it is unclear that we should equate ‘metacognitive error’ with ‘metacognitive disorder’. After all, metacognitive errors are the norm: most people are overconfident of their abilities, and almost everyone thinks that they are better drivers than average.

Relatedly, as many as 50% of people with a diagnosis of FCD have co-morbidity with psychological disorders such as depression (e.g., Bhome, 2019) which renders the idea FCD as a primary disorder in metacognition as somewhat suspect.

END

Author Response

Reviewer 2:

This article raises the intriguing hypothesis that ‘Functional Cognitive Disorder’ is ‘primary disorder of metacognition’ (p. 4, line 149). As such, this is an interesting hypothesis. However, I admit I found the manuscript hard to follow. It felt that the author tried to cover too much, and the end-result is a product that is too broad and not sufficiently in-depth. I had a hard time noticing how the following attempt advances our understanding beyond Bhome et al (2019)’s review on the same topic. I’m sorry I do not have a more positive disposition toward it. Below I list some observation that might be useful to the author.

As described by the authors, ‘Functional Cognitive Disorder’ refers to a subjective deficit in cognition in the context of objectively normal performance. An illustrative example would be a person who seeks the services of a memory clinic or a cognitive neurology clinic despite having a normal neuropsychological profile.

First, I want to raise as an important caveat that the gap between subjective and objective deficit does not necessarily mean that the person has a deficit (in metacognitive) judgment. Instead, the patient might very well be impaired relative to his/her baseline and the cognitive battery fail to detect such a decline, either because the patients’ baseline was unusually high, or because the objective test is less sensitive than the patient’s insight.

Author response: The risk of misdiagnosis of early-stage neurodegeneration as FCD has been mentioned and further developed (Section 1, paragraph 5).

Other additional criteria have been proposed to better describe Functional Cognitive Disorders. While the author does cite such work (e.g., by McWhither et al; Stone et al, 2015, etc.) as a reader I had a hard time getting a solid grasp of the diagnosis criteria from the text. . For example, it was unclear whether the diagnosis requires a failure to progress (so to differentiate it from dementias). The author would be well advised to start the article with a clear definition of FCD, such as “Functional cognitive disorder (FCD) is characterised by the experience of persistent and distressing subjective cognitive difficulties in the absence of detectable objective cognitive deficit and underlying brain pathology” (Bhome et al., 2019)

Author response: Whilst starting with a clear definition is a legitimate approach, as per Bhome et al., it might be argued that outlining some of the key empirical observations which helped to build to a consensus operational definition is more reflective of the evolution of the field.  The current consensus definition of FCD has been given (Box 1), which encompasses the points in the definition suggested by Bhome and colleagues.

Putting these issues aside, it is definitional that people who exhibit a gap between   objective performance and the subjective self-assessment, are committing a metacognitive error. However, it is unclear that we should equate ‘metacognitive error’ with ‘metacognitive disorder’. After all, metacognitive errors are the norm: most people are overconfident of their abilities, and almost everyone thinks that they are better drivers than average.

Author response: This distinction has now been addressed (Section 2, paragraph 3).

Relatedly, as many as 50% of people with a diagnosis of FCD have co-morbidity with psychological disorders such as depression (e.g., Bhome, 2019) which renders the idea FCD as a primary disorder in metacognition as somewhat suspect.

Author response: In light of this comment, the possible contribution of metacognitive mechanisms in the occurrence of depression has been added (Section 5.3, paragraph 5) and referenced.

Reviewer 3 Report

The main goal of the current study was to shed light on the role of metacognition in functional cognitive disorders (FCD) at a theoretical level taking into account the existing empirical data. The researchers focused on the hypothesis that FCD are possibly disorders of metacognition, most usually of metamemory. After reviewing previous adaptations of theoretical models of functional neurological disorders to accommodate FCD, they attempted to approach FCD in the light of relevant and influential metacognitive models.

This is an interesting study with valid research questions, adequately researched, analytical and informational. It applies correct and transparent methodology. The figures as well the tables are very useful. The English language is appropriate and understandable. Among the most interesting points, we can mention that the researchers ask a critical question proposing an innovative approach.

 As regards the limitations of this paper, we consider that the article has an obvious lack of references to relevant works like the following suggested, and this is a handicap of this article that should be corrected, in order for its study and review of the field to be considered completed. We are of the opinion that the researchers should take into account additional models that support their research idea. In the section that the researchers examine the possible relation between metacognitive models and FCD, we recommend the inclusion of Drigas’ et al.'s extensive work. Drigas and Pappas (2017) proposed the 8*8 layered model of human cognition, which adapts to the different types of intelligence and operates under the rules of metacognitive procedures, such as monitoring, regulation and adaptation. Based on the aforementioned layered model, Drigas and Mitsea (2020) presented the 8 pillars model of metacognition which describes the core components of metacognition shedding light on the underlying neuropsychological mechanisms. According to this model, metacognition is a prerequisite for self-awareness, for being conscious about the strengths and the limits of our own cognition, for observing and recognizing when and how our cognitive functions malfunction. In addition, they conclude that attention and memory operate simultaneously as cognitive and metacognitive abilities, in order to support monitoring and control processes. Thus we think, that the aforementioned models go hand in hand with the needs of the current research.

In their conclusions, the authors point out the importance of metacognitive training in order to improve their metamemory.

It is an interesting point that the researcher should emphasize with a brief reference on Drigas and Mitsea’s (2019, 2021) research who studied a wide range of metacognitive strategies in order to cope with the disorders of metacognition.

Thus, we recommend the inclusion of the following papers:

  1. Drigas, A. S., & Pappas, M. A. (2017). The consciousness-intelligence-knowledge pyramid: an 8x8 layer model.International Journal of Recent Contributions from Engineering, Science & IT (iJES),5(3), 14-25.
  2. Drigas, A., & Mitsea, E. (2020). The 8 pillars of Metacognition.International Journal of Emerging Technologies in Learning (iJET),15(21), 162-178.
  3. Drigas, A., & Mitsea, E. (2021). Metacognition, Stress-Relaxation Balance & Related Hormones. J. Recent Contributions Eng. Sci. IT9(1), 4-16.
  4. Mitsea, E., & Drigas, A. (2019). A Journey into the Metacognitive Learning Strategies.International Journal of Online & Biomedical Engineering15(14).

We believe that this paper after the suggested improvements could positively contribute in the international literature to the domain of dementia and metacognition.
